# Children and Adolescents with Pulmonary Arterial Hypertension: Baseline and Follow-Up Data from the Polish Registry of Pulmonary Hypertension (BNP-PL)

**DOI:** 10.3390/jcm9061717

**Published:** 2020-06-03

**Authors:** Joanna Kwiatkowska, Malgorzata Zuk, Anna Migdal, Jacek Kusa, Elzbieta Skiba, Karolina Zygielo, Kinga Przetocka, Piotr Werynski, Pawel Banaszak, Alina Rzeznik-Bieniaszewska, Rafal Surmacz, Waldemar Bobkowski, Barbara Wojcicka-Urbanska, Bozena Werner, Joanna Pluzanska, Katarzyna Ostrowska, Anna Waldoch, Grzegorz Kopec

**Affiliations:** 1Department of Pediatric Cardiology and Congenital Heart Defect, Medical University of Gdansk, 80-210 Gdansk, Poland; joannak@gumed.edu.pl (J.K.); anna.waldoch@gumed.edu.pl (A.W.); 2The Children’s Memorial Health Institute, 04-730 Warsaw, Poland; zukmala@gmail.com (M.Z.); a.migdal@op.pl (A.M.); 3Pediatric Cardiology Department, Regional Specialist Hospital- Research and Development Centre in Wroclaw, 51-124 Wroclaw, Poland; jkusa@poczta.onet.pl (J.K.); elzskiba@wp.pl (E.S.); kzygielo@gmail.com (K.Z.); 4Department of Pediatric Cardiology, Jagiellonian University Medical College, 30-663 Krakow, Poland; kprzetocka@usdk.pl (K.P.); piotr.werynski@uj.edu.pl (P.W.); 5Department of Congenital Heart Disease and Pediatric Cardiology, Silesian Centre for Heart Diseases, 41-800 Zabrze, Poland; rhplus@op.pl; 6Department of Pediatric Cardiology, Poznan University of Medical Sciences, 60-572 Poznan, Poland; alina.bieniaszewska@gmail.com (A.R.-B.); rsurmacz@mp.pl (R.S.); wbobk@mp.pl (W.B.); 7Department of Pediatric Cardiology and General Pediatrics, Medical University of Warsaw, 02-091 Warsaw, Poland; barbara.wojcicka@wum.edu.pl (B.W.-U.); bozena.werner@wum.edu.pl (B.W.); 8Cardiology Department, Polish Mother’s Memorial Hospital-Research Institute, 93-338 Lodz, Poland; asia.pluzanska@yahoo.com (J.P.); kostrowska9@wp.pl (K.O.); 9Department of Cardiac and Vascular Diseases, Jagiellonian University Medical College, John Paul II Hospital in Krakow, 31-202 Krakow, Poland

**Keywords:** pulmonary hypertension, pediatric registry, epidemiology

## Abstract

We present the results from the pediatric arm of the Polish Registry of Pulmonary Hypertension. We prospectively enrolled all pulmonary arterial hypertension (PAH) patients, between the ages of 3 months and 18 years, who had been under the care of each PAH center in Poland between 1 March 2018 and 30 September 2018. The mean prevalence of PAH was 11.6 per million, and the estimated incidence rate was 2.4 per million/year, but it was geographically heterogeneous. Among 80 enrolled children (females, *n* = 40; 50%), 54 (67.5%) had PAH associated with congenital heart disease (CHD-PAH), 25 (31.25%) had idiopathic PAH (IPAH), and 1 (1.25%) had portopulmonary PAH. At the time of enrolment, 31% of the patients had significant impairment of physical capacity (WHO-FC III). The most frequent comorbidities included shortage of growth (*n* = 20; 25%), mental retardation (*n* = 32; 40%), hypothyroidism (*n* = 19; 23.8%) and Down syndrome (*n* = 24; 30%). The majority of children were treated with PAH-specific medications, but only half of them with double combination therapy, which improved after changing the reimbursement policy. The underrepresentation of PAH classes other than IPAH and CHD-PAH, and the geographically heterogeneous distribution of PAH prevalence, indicate the need for building awareness of PAH among pediatricians, while a frequent coexistence of PAH with other comorbidities calls for a multidisciplinary approach to the management of PAH children.

## 1. Introduction

Pulmonary arterial hypertension (PAH) is a rare disease characterized by rapid progression and poor prognosis. In the pediatric population, its incidence and prevalence have been estimated at 4–10 cases per million children per year, and 20–40 cases per million children, respectively [1]. Prior to the availability of the targeted PAH therapies, the estimated median survival of children with idiopathic pulmonary arterial hypertension (IPAH) was between 10 months and 4.1 years [2,3]. However, it improved significantly with the advent of targeted treatment [4,5,6,7].

Current epidemiologic data on PAH in children comes from the registries of the western European and the US populations [8,9,10,11,12]. However, several factors which affect the prevalence, natural history and outcomes of pediatric PAH—such as lifestyle and environmental factors, degree of access to modern diagnostic tests and interventional procedures, treatment standards and awareness of the disease among health care providers—are inhomogeneously distributed throughout the world. Consequently, data from one geographical region cannot be easily translated to other parts of the world. 

The DataBase of Pulmonary Hypertension in the Polish Population (Baza Nadciśnienia Płucnego; BNP-PL, https://clinicaltrials.gov/ct2/show/NCT03959748) is the first multicenter prospective registry of adult and pediatric PAH and chronic thromboembolic pulmonary hypertension (CTEPH) patients created in any central-eastern European country [13]. In the present study, we aimed to characterize the Polish PAH patients aged under 18 years in the era of modern PAH therapy. Specifically, we aimed to assess the prevalence of PAH patients in this population, the typical disease presentation, diagnostic procedures, methods of treatment, and also the associated morbidity. These data should provide useful information to guide future clinical management of pediatric PAH patients.

## 2. Methods

### 2.1. Design of the BNP-PL Registry—PAH Children Arm

The design of the BNP-PL registry and enrolment criteria have been recently presented in detail [13]. Herein, we present only the features of our study which are distinctive to the pediatric population (PAH—Children arm of the BNP-PL registry). In Poland, the management of children with PAH is centralized in eight reference cardiological centers, accredited by the National Health Fund to treat PAH. All of the centers have been participating in our registry. For the present study, we prospectively enrolled PAH patients, at ages of three months to 18 years, who had been under the care of one of the PH (Pulomonary Hypertension) centers between 1 March 2018 and 30 September 2018. Patients diagnosed before 1 March 2018 were defined as prevalent cases, while patients diagnosed on 1 March 2018 and later were defined as incident cases. 

The protocol of the study was reviewed and accepted by the Bioethical Committee of Physicians and Dentists Chamber in Krakow (L.dz.OIL/KBL/27/2018).

### 2.2. Diagnostics

To be enrolled in the study, all patients had to undergo diagnostic right heart catheterization (RHC), confirming the PAH diagnosis based on the European Society of Cardiology’s definition [14]: mean pulmonary arterial pressure (mPAP) of 25 mm Hg or more, mean pulmonary arterial wedge pressure (PAWP) or left ventricular end-diastolic pressure (LVEDP) of 15 mm Hg or less, pulmonary vascular resistance index (PVRI) of > 3 Woods Units (WU) and a clinical diagnosis of PAH according to the clinical investigators’ judgments, guided by generally accepted definitions. Patients were classified by the investigators, based on the clinical classification of PAH [1,14] as recommended, into idiopathic (IPAH) or associated with connective tissue disease (CTD-PAH), congenital heart disease (CHD-PAH), portal hypertension (Po-PAH), drugs or toxins (DPAH), HIV infection (HIV-PAH) or schistosomiasis. As genetic testing was not available in most centers, we did not distinguish a separate class for patients with heritable PAH. Patients with CHD-PAH were further sub-classified as Eisenmenger’s syndrome, PAH associated with prevalent systemic-to-pulmonary shunts, PAH with small/coincidental defects, or PAH after the defect correction [15].

The acute pulmonary vasoreactivity test (AVT) for patients with idiopathic PAH was performed. A positive response to AVT in this group of children was defined according to modified Barst or Sitbon [16] criteria, depending on the center. The following vasodilators were used in AVT: nitric oxide (*n* = 20), iloprost (*n* = 1), adenosine (*n* = 2), epoprostenol (*n* = 1), oxygen (*n* = 1). The long-term response to calcium channel blockers was defined as WHO (World Health Organization) Functional Class I/II with a sustained hemodynamic improvement (same or better than achieved in the acute test) after at least one year of being treated with calcium channel blockers only.

The diagnostic work-up included echocardiography and clinical status assessment [anthropometric data, WHO functional class, 6 min walk test and *n*-terminal pro brain natriuretic peptide (NTproBNP) when available], and additionally an exclusion of chronic thromboembolic disease and pulmonary disease. Body mass and height were expressed in kg and cm respectively, and also as percentiles. Body mass index (BMI) was calculated as body mass[kg]/height[m^2^]. Significant growth failure was defined as height below the third percentile. We used percentile charts for growth based on the data obtained for the Polish population [17,18]. As the BNP-PL registry is entirely observational, the protocol of the study does not require additional patient visits or diagnostic tests and has had no influence on the management of patients.

### 2.3. Treatment

In Poland, children with PAH are treated based on the ESC (European Society of Cardiology) guidelines [14] and AEPC 2019 updated consensus statement [19], adjusted to the National Health Fund criteria for reimbursement of PAH-targeted therapies. At the time of enrolment, monotherapy (sildenafil or bosentan) or dual oral combination therapy (sildenafil plus bosentan) were reimbursed, however some patients could have been treated with prostacyclin analogues by individual donation. The changes in the reimbursement policy from 1 November 2018 allowed the use of a triple combination therapy composed of sildenafil, bosentan and prostacyclin analogue (treprostinil, epoprostenol or iloprost in inhalations). Calcium channel antagonists were reserved for patients with idiopathic PAH with positive AVT. Additional therapies could have been used in patients enrolled in clinical trials, which was also recorded in our database. Patients were also analyzed with respect to other therapies, including: home oxygen therapy, vitamin K antagonists, low-molecular weight heparin, new oral anticoagulants, beta blockers, angiotensin converting enzyme inhibitors, angiotensin receptor blockers, ivabradine, amiodarone, loop diuretics, thiazide diuretics, potassium-sparing diuretics, selective serotonin reuptake inhibitors, acetylsalicylic acid, proton pump inhibitors, statins, corticosteroids and immunosuppressive therapy.

### 2.4. Prevalence, Incidence and Follow-Up

In the present study, we calculated the period prevalence of PAH. The numerator included all patients diagnosed with PAH in every center who were alive on 1 March 2018, and all new patients diagnosed between 1 March 2018 and 30 September 2018. The denominator was the number of Polish children and adolescents (<18 years old) in Poland (*n* = 6920652) based on the most recent data (31 December 2017) from Statistics Poland (Central Statistical Office), which is Poland’s chief government executive agency in charge of collecting and publishing statistics related to the country’s economy, population and society, at the national and local levels.

The incidence was calculated as an estimated number of new PAH cases per year (numerator) per the number of children and adolescents (<18 years old) in Poland (denominator; *n* = 6920652), based on the most recent data (31 December 2017) from Statistics Poland. The estimated number of new PAH cases per year was the product of 12 months and the mean number of new cases per month. 

For the purposes of the present study, the follow-up observation was carried out from the time of enrolment until 31 August 2019. As previously described [13,20], several parameters were recorded, including deaths, hospitalizations and changes in treatment.

### 2.5. Statistical Methods

Continuous variables were reported as median (interquartile range). Categorical variables were reported as counts and percentages. For the comparison of continuous variables between two groups, we used the U Mann–Whitney test, and for categorical variables, the χ2 test, with Yate’s correction as needed. Kaplan–Meier survival curve was used to delineate patient’s survival.

The significance level was set at alpha level of 0.05.

Statistical analysis was performed with the use of Dell Inc. (2016), Dell Statistica (data analysis software system), version 13 (Dell, Texas, TX, USA, software.dell.com.).

## 3. Results

### 3.1. Study Group

A total of 87 children with PH from eight reference centers in Poland entered the electronic database between 1 March 2018 and 30 September 2018. Two patients were identified to be duplicated, and the duplicate entries were discarded. Five other children were identified to belong to Group II, III or V, and they were also discarded. Finally, 80 PAH children met the study entry criteria, and these patients were enrolled in the present analysis. A total of 10 (12.5%) patients were diagnosed de novo between 1 March 2018 and 30 September 2018, and they were referred to as incident cases. The other 70 (87.5%) patients were diagnosed before 1 March 2018, and they were referred to as prevalent cases.

### 3.2. Prevalence, Incidence and Geographic Distribution of PAH

The mean prevalence of PAH was 11.6 per million children (11.3 per million in males and 11.9 per million in females), and the estimated incidence rate was 2.4 per million children per year. CHD-PAH was more prevalent than IPAH (7.8 per million vs. 3.6 per million). The geographical distribution of PAH was heterogeneous, ranging from 0 to 24.3 per million children in different regions (Figure 1).

### 3.3. PAH Classification

As shown in Figure 2, 25 patients were classified as idiopathic (IPAH), 54 as PAH associated with congenital heart disease (CHD-PAH) and one child with portopulmonary hypertension (Po-PAH). We have not identified any patient with other forms of PAH. In the IPAH group, four patients had positive AVT at the time of diagnosis. Two of them were still vasoreactive at the time of enrolment. In the CHD-PAH group, the most frequent diagnosis was an Eisenmenger’s syndrome (*n* = 16; 29.6%). In Table 1 we show the prevalence of different CHD defects according to the clinical classification of CHD-PAH. In 16 (29.6%) patients, we found complex congenital heart lesions.

### 3.4. Concomitant Diseases in IPAH and CHD-PAH 

As presented in Table 2, 33 (40%) patients exhibited mental retardation, among them 24 (73%) had Down’s syndrome and 9 (27%) had other genetic diseases, such as Noonan syndrome (*n* = 2), DiGeorge syndrome (*n* = 2), Alagille syndrome (*n* = 1), Moyamoya disease (*n* = 1), complex structural chromosomal aberration [duplication of 11 p15.5 p15.2 region with concomitant deletion of 11 q24.3 q25 region (*n* = 1)] and chromosomal aberration [deletion of 13 p (*n* = 1), chromosome nine trisomy (*n* = 1)]. Hypothyroidism, mental retardation and Down’s syndrome were found more frequently in patients with CHD-PAH than in patients with IPAH.

### 3.5. Demographics of Study Patients

All patients were of Caucasian origin. As shown in Table 3, males and females were represented in similar proportions, and were at similar ages at PAH diagnosis and at the time of enrolment in the study. Prevalent patients were older than incident cases [median 11.5 (8.6–15.4) vs. 5.4 (1.7–8.6) years, respectively, *p* = 0.002], but the age of PAH diagnosis was similar in the incident (5.7; 0.8–8.3) and prevalent cases (5.1; 2.1–7.6), *p* = 0.9. Body weight at the time of enrolment ranged from 5.3 to 77.6 median 30.3 kg (20.0–49.4), and height ranged from 0.62 to 1.89 median 1.34 m (1.14–1.53). The distribution of percentiles of body weight, height and BMI is presented in Figure 3. The percentiles of median height, weight and BMI for age are shown in Table 3. Patients with genetic defects, as compared to patients without genetic defects (Table 2), presented decreased median height for age percentile (*p* = 0.006) and median weight for age percentile (*p* = 0.009), and similar BMI for age percentile (*p* = 0.3). The shortage of growth was found in 16 (30%) patients with CHD-PAH and in 4 (16%) patients with IPAH (*p* = 0.2). All patients attended school or preschool, according to their age.

### 3.6. Clinical Presentation at Diagnosis

Among the incident patients, the most commonly presenting symptom was fatigue, which was reported by 9 (90%) children or their parents; it was followed by exercise dyspnea [reported by 7 (70%) subjects] and resting dyspnea [reported by 2 (20%) subjects]. The other symptoms, including chest pain, cough, syncope or presyncope and heart palpitations, were each reported once.

At the time of PAH diagnosis, 9 children were at FC I, 17 at FC II, 53 at FC III and 1 at FC IV. For the incident cases, the delay from the onset of the first symptoms to the diagnosis was 0.3 (0.1–0.7) years.

### 3.7. Diagnostic Work-up

The diagnosis of pulmonary hypertension was confirmed by RHC in all study patients (inclusion criterion). Postcapillary pulmonary hypertension was excluded based on echocardiography and measurement of PAWP during RHC.

To exclude chronic thromboembolic disease, 49 (61%) patients underwent pulmonary arteriography, and 6 (7.5%) patients underwent computed tomography angiography. Ventilation- perfusion scans were not performed in the study group. In 7 patients diagnosed as IPAH, and in 17 patients with CHD-PAH, the exclusion of chronic thromboembolism was based only on the clinical assessment. Significant pulmonary diseases in most cases were excluded based on the medical history, or the physical examination and the chest X-ray. Additional spirometry was required in 4 patients and high resolution computed tomography in 14 patients. The results of the 6 min walk test (6MWT) were available in 50 patients, and the right atrial area (RAA) in 29 patients, at the time of enrolment.

### 3.8. Clinical Characteristics of Study Group

At the time of enrolment, 10 (12.5%) children were ranked as FC I, 45 (56.3%) as FC II and 25 (31.3%) as FC III. The median 6 MWTwas 429 (360–500) m. The baseline characteristics of the study patients are presented in Table 3. There were no significant differences with regard to the functional capacity, 6 MWT, NT-proBNP level or hemodynamics between the groups of CHD-PAH and IPAH patients at the time of enrolment.

### 3.9. Treatment

PAH-specific treatments among IPAH and CHD-PAH patients at the time of enrolment are shown in Table 4. All patients received targeted PAH therapies, except 2 (2.5%) patients with reactive PAH who received only calcium channel blockers for their disease. A total of 36 (45%) patients were treated with monotherapy, 39 (48.8%) with double oral therapy, and 3 (3.8%) with a combination of oral drugs and prostacyclin analogues (triple combination therapy). One patient with the diagnosis of portopulmonary PAH was treated with sildenafil as monotherapy. Endothelin receptor antagonist (bosentan) was used by 61 (76.8%) patients. The group of phosphodiesterase-5 inhibitors was represented by sildenafil, which was used by 57 (71.3%) patients, and tadalafil was used by 1 (1.3%) patient. Prostacyclin analogues were represented only by subcutaneous treprostinil, which was used by three patients. One patient used riociguat, being a participant in the open label extension phase of a clinical trial at the time of enrolment.

In total, as shown in Table 4, 51 (63.8%) children at the time of enrolment received diuretics, and 18 (22.5%) were treated with angiotensin-converting enzyme (ACE)-inhibitors or angiotensin receptor blockers (ARBs). In addition, 3 (3.8%) patients were treated with digoxin and 6 (7.5%) with beta-blockers. Oral anticoagulation in the form of vitamin K antagonists was administered to 12 (15%) of the patients and aspirin to 10 (12.5%). The other analyzed groups of drugs, as presented in the Methods section, were not used. There were no significant differences with regard to medication between CHD-PAH and IPAH groups, except potassium-sparing diuretics (*p* = 0.02).

### 3.10. Follow-Up

During the follow-up observation period of 16.5 ± 2.1 months, two (one IPAH and one CHD-PAH) patients died at home due to the progression of PAH (Figure 4). Before death, they were treated with sildenafil and sildenafil plus bosentan, respectively, and both were ranked as WHO FC III during their last outpatient visits. Additional nine patients (three patients with IPAH, six patients with CHD-PAH) were hospitalized; two patients for diagnostic reasons, one patient for an additional intervention (reimplantation of pulmonary trunk prosthesis and balloon angioplasty of right PA), two patients due to respiratory infections and five patients for the escalation of PAH therapy. All the hospitalized patients were discharged home. PAH-specific therapies were changed in 11 patients (4 patients with IPAH and 7 patients with CHD-PAH). During the follow-up, the triple combination therapy was used in an additional seven patients: sildenafil, bosentan and treprostinil in six patients, and sildenafil, bosentan and epoprostenol in one patient. In five patients, monotherapy with bosentan or sildenafil was changed to the double oral combination therapy with bosentan and sildenafil. One patient with CHD-PAH, who was treatment naïve at the time of enrolment, was administered the treatment with macitentan during the follow-up (open label clinical trial).

## 4. Discussion

The present analysis of the BNP-PL registry shows for the first time the demographics, treatment and the burden of coexisting diseases for a group of Caucasian children with PAH of central-eastern European origin.

The main features of this population include a high proportion of patients with CHD-PAH, as compared to patients with IPAH, and underrepresentation of other forms of PAH, a heterogenous geographical distribution of patients, the coexistence of important comorbidities, and the need for improving the treatment.

The finding of the heterogenous regional distribution of PAH was largely unexpected by the authors, since populations in different Polish areas are homogenous. We were not able to explain this observation via analysis of the access to reference centers, as all regions with less than median PAH prevalence had at least one pediatric PAH center, and four regions with the highest prevalence had no pediatric PAH center. Accordingly, the authors did not identify any specific reason for the regional heterogenity of PAH prevalence. However, similarly to other authors who made the same observation in their populations [20,21], we consider that the awareness of PAH is insufficient among pediatricians, and consequently many patients may not be identified or referred to specialized centers.

The mean prevalence of PAH in Poland, according to our registry, is 11.6 per million children. Registries from other countries show a similar prevalence of IPAH, but the prevalence of CHD-PAH is rarely reported. In Table 5, we summarize the results of other PAH registries. 

### 4.1. PAH Classification

In our study, in most patients, PAH was associated with CHD, which occurred twice as often as IPAH. The proportion of these two most-frequent diagnoses of PAH (CHD-PAH and IPAH) was similar in the Dutch PAH registry [11] and in the Spanish one [22]. In other reports, IPAH was the dominant form of PAH in children [4,23]. In the TOPP (Tracking Outcomes and Practice in Pediatric Pulmonary Hypertension) registry, which enrolled PH patients from 31 pediatric centers in 19 countries, PAH was the most frequent type of PH, accounting for 317 (88%) out of 362 children. In the PAH group, most patients were diagnosed with IPAH/HPAH (*n* = 243; 57%), and this was followed by patients with CHD-PAH (*n* = 161; 38%) and other forms of APAH (*n* = 23; 4%) [23]. Similarly, the REVEAL registry, a multicenter study conducted in the United States, showed that IPAH/HPAH and CHD-PAH, when combined, accounted for most (*n* = 199; 92%) diagnoses of PAH; IPAH/HPAH was diagnosed in 122 (56%) patients and CHD-PAH in 77 (36%) patients [10]. We speculate that the dominance of CHD-PAH over IPAH in our population might be attributable to the longer survival of patients with PAH, due to cardiac defects, than IPAH patients [24], however, the exact cause of the CHD-PAH dominance is not known. In the present study, we found only one patient with PoPAH (a condition rarely diagnosed in children [25]), and no patient was diagnosed with other forms of associated PAH, including CTD-PAH. Although these types of PAH were also very rare in other registry studies 10 (4.6%) patients with CTD-PAH and three (1.4%) patients with PoPAH in the REVEAL registry [4], our data suggest the need for building an awareness of PAH in pediatricians specializing in children rheumatology and hepatology.

In the group of IPAH, 16% of patients had a positive acute vasoreactivity test at diagnosis, but at the time of enrolment, only half of them were still considered vasoreactive. Other pediatric registries reported positive results of acute vasoreactivity testing ranging from 6% to 34%, but the comparison of different studies is difficult due to the various protocols of the test [4,6,7,12,26,27]. Still, it is important to check the acute pulmonary vasculature response to vasodilators, since as many as half of the patients with a positive test can be successfully treated with calcium channel blockers in the long-term.

As genetic testing for PAH-specific mutations is not routinely performed in Poland, we were not able to identify patients with a diagnosis of HPAH. Therefore, it must be acknowledged that in some patients originally referred as IPAH, genetic screening would have corrected the diagnosis to HPAH.

### 4.2. Growth in Children with PAH

In the present study, we showed that a significant number of children with PAH had growth impairment. For the assessment of growth, we referred our data to the values of weight, height and BMI, obtained for the general population of Polish children and adolescents [17,18]. Growth retardation was most frequent in patients with Down syndrome or other genetic defects, but it also occurred in patients without additional comorbidities. It was also more frequent in CHD-PAH than in IPAH patients. Importantly, a significant number of patients (CHD-PAH 30%, IPAH 16%) showed a severe growth retardation (height below third percentile for age). Our results are in line with the data from the recently published longitudinal retrospective multi-registry study. Ploegstra et al. [28] showed that PAH was associated with growth impairment, and also that its degree correlated with the disease severity and duration of PAH. They also observed that a favorable clinical course of PAH was associated with catch-up growth, and concluded that height-for-age could serve as an additional and easily available clinical parameter to monitor patients’ clinical condition. In addition, the data from the UK national cohort study [29] suggest that growth retardation is connected with worse survival for children with IPAH. Based on our data, we conclude that there is a need for strict cooperation between PH physicians, endocrinologists and dietary specialists in the course of treatment of PAH children.

**Table 5 jcm-09-01717-t005:** Review of pediatric pulmonary hypertension registries.

Author	Publication Year	*N*	Incidence (*N* per Million Children)		Prevalance (*N* per Milion Children)		Diagnosis (%)			Symptoms (%)					Down Syndrome (%)
			IPAH	PAH-CHD	IPAH	PAH-CHD	IPAH	PAH-CHD	Other PAH	Fatigue	Excercise Dyspnea	Dyspnea	Chest pain	Syncope	
Moledina [29]	2010	64	0.48		2.1		IPAH only			31	31	75		31	4.7
Fraisse [12]	2010	50			2.2		60	24	16	72	76		7	11	
van Loon [11]	2011	154	0.7	2.2	4.4	15.6	23	72	5		70	43	1	7	18
Barst [4]	2012	216					56	36	8	24	46	9	5	24	
del Cerro [22]	2014	142	0.49		2.9		21	74	5		76		6.2	9.3	17
Kwiatkowska	2020	80			3.6	7.8	31	67	2	90	70	20	1.25	1.25	30

IPAH, idiopathic pulmonary arterial hypertension; PAH-CHD, pulmonary arterial hypertension associated with congenital heart disease

### 4.3. PAH-Targeted Therapies

Our patients were usually treated with endothelin receptor antagonists and PDE-5 inhibitors. The use of prostacyclin analogues was very low. Only half of our group used a combination of two PAH-targeted treatments, which in the majority of patients were composed of sildenafil and bosentan. The low use of prostacyclin analogues and combination therapy could have been at least partly attributed to the rigorous reimbursement policy in Poland. Before November 2018, prostacyclin analogues, and consequently the triple combination therapy, had not been reimbursed for children with PAH in Poland. A change in the reimbursement policy led to a 31% increase in the use of combination therapies, and more than a doubling of the use of the triple combination therapy during the one-year observation time.

Although guidelines on PAH treatment in children follow the recommendations for adults, advocating an early combination of different PAH-targeting therapies [14], the evidence for treatment protocols in children is still very limited [30,31]. Accordingly, the use of combination therapies in children may still be inadequately low. The data from a large population of PAH patients aged under 18 years, treated in the US between 2010 and 2013, showed that double and triple combination therapies were used in 13.1% and 3.2% of patients, respectively [10]. In the analysis of incident PAH patients enrolled in the multinational TOPP registry between 2008 and 2012, at the time of diagnosis, the PAH-targeted therapy administered to most patients was monotherapy, and the double and triple combination therapies were applied to 18% and 2.3% of patients, respectively [32]. The analysis of more recent data from the Netherlands (June 2013–March 2016) shows an increased use of double and triple combination therapies in PAH children [15 (50%) and five (17%), respectively] [33,34]. Our registry shows a significant influence of the reimbursement policy on treatment scenarios in PAH children. The proportion of patients using the triple combination therapy increased from 3.8%, when the triple combination was not reimbursed, to 13% in less than a year, after the change in the reimbursement policy.

At the time of study recruitment and follow-up, no patient was listed for transplantation.

### 4.4. Comorbidities 

In the present study, PAH was most frequently associated with mental retardation, Down syndrome and other genetic disorders. A significant number of patients suffered from hypothyroidism. These comorbidities were far more prevalent in the CHD-PAH group than in IPAH patients. The high proportion of genetic disorders among CHD-PAH patients is in line with the data from other registries.

For example, in a recently-performed multicenter study of cohort of CHD-PAH children, genetic abnormalities were described in 29% of patients [15].

Of note was a high prevalence of hypothyroidism in our PAH population, much higher than in the population (2.7%) of PAH children treated in the US [10]. Our results might have been attributed to the large representation of patients with Down syndrome, however, even in the group of patients with Down syndrome, hypothyroidism was far more frequent than previously reported. The coexistence of genetic disorders, mental retardation, hormonal abnormalities and growth retardation in children with PAH indicates the need for multi-specialty care in this group of patients [35].

### 4.5. Strengths and Limitations

Our study has several strengths. First of all, our study enrolled an unbiased group of exclusively Polish children with PAH. Secondly, we are the first to show the multicenter, prospectively collected data on PAH epidemiology in children in a central-eastern European country with a significantly different political and economic background than western countries. Thirdly, we have showed how the reimbursement policy impacts therapeutic decisions in PAH.

Our study also has some limitations. The first is the relatively small number of newly diagnosed patients as compared to the previously-diagnosed ones. Secondly, we have not identified monogenic forms of PAH, as genetic testing has not been routinely performed in Polish patients. Thirdly, the group of patients with other forms of associated PAH, especially CTD-PAH, seems to be underrepresented in our registry, which indicates the need for building the awareness of PAH among Polish pediatricians.

## 5. Conclusions

Our study demonstrates that almost all PAH cases were associated with CHD, or were diagnosed as IPAH without a significant representation of other PAH subclasses. Additionally, we found a heterogeneous geographic distribution of PAH prevalence. These two observations may indicate that some children are underdiagnosed, and that building the awareness of PAH is mandatory among pediatricians. We have also shown that PAH in children coexists with important comorbidities, which calls for a multidisciplinary approach to the management of this disease. We have also concluded that building a national strategy for PAH treatment in children, with the reimbursement policy adapted to current medical guidelines, improves the PAH patients’ chance of receiving the optimal treatment.

## Figures and Tables

**Figure 1 jcm-09-01717-f001:**
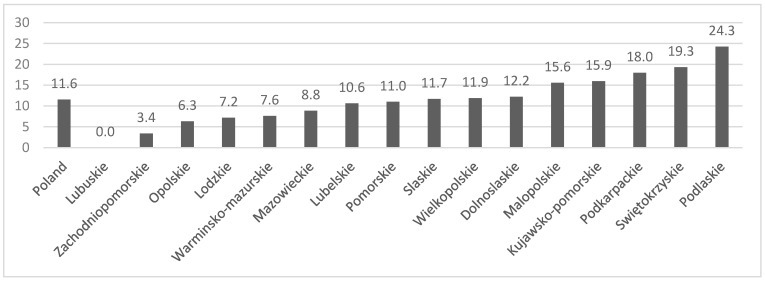
Geographical distribution of patients with pulmonary arterial hypertension (PAH). The mean prevalence of PAH per million children in different Polish voivodeships.

**Figure 2 jcm-09-01717-f002:**
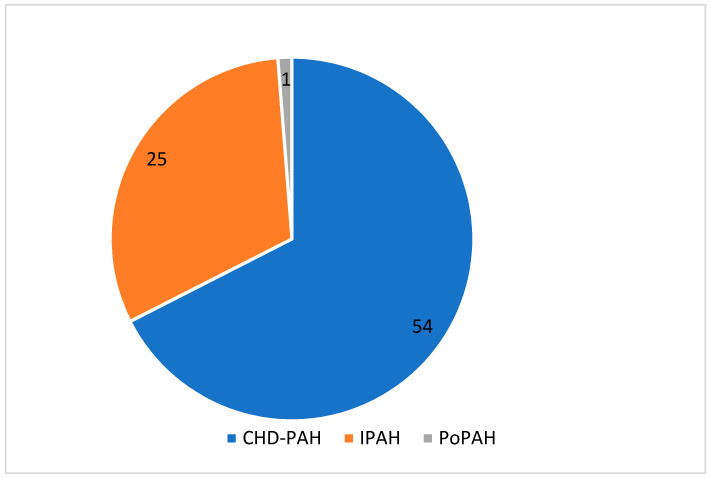
Distribution of different classes of pulmonary arterial hypertension in the study group. CHD-PAH, pulmonary arterial hypertension associated with congenital heart disease; IPAH, idiopathic PAH; PoPAH, portopulmonary PAH.

**Figure 3 jcm-09-01717-f003:**
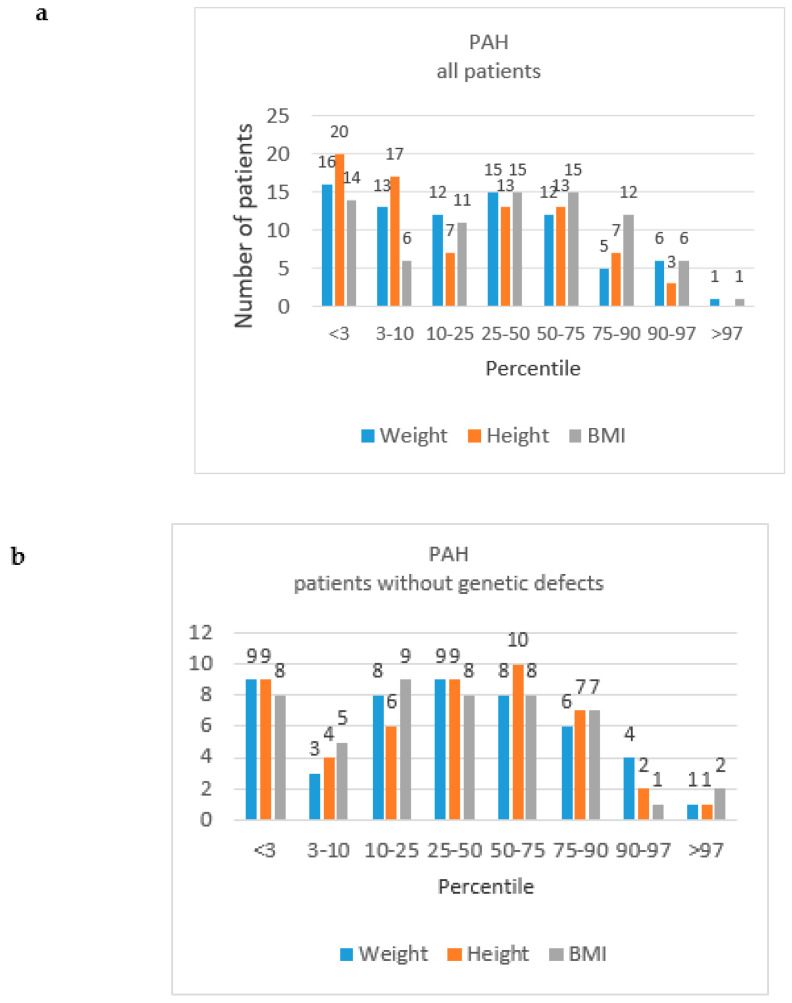
The distribution of the number of patients with PAH by weight, height and body mass index (BMI) percentiles, by “all patients” study groups and “patients without genetic defects” groups. (**a**) PAH, all patients; (**b**) PAH, patients without genetic defects; (**c**) IPAH, all patients; (**d**) IPAH, patients without genetic defects; (**e**) CDH-PAH, all patients; (**f**) CDH-PAH, patients without genetic defects.

**Figure 4 jcm-09-01717-f004:**
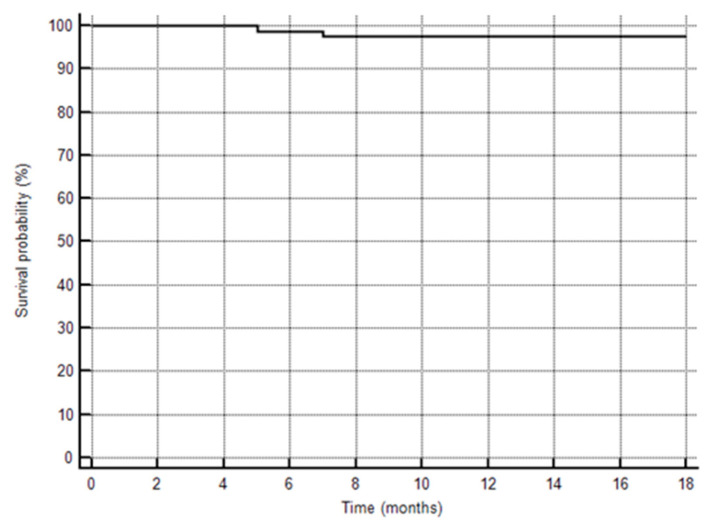
Kaplan–Meier curve to show survival of the study patients.

**Table 1 jcm-09-01717-t001:** The prevalence of different heart defects in four subgroups of patients (CHD-PAH: *n* = 54) according to the European Society of Cardiology.

	Eisenmenger’s Syndrome	PAH Associated with Prevalent Systemic-to- Pulmonary Shunts	PAH after Defect Correction	PAH with Small/Coincidental Defects
	*N* = 16	*N* = 15	*N* = 15	*N* = 8
ASD, *n* = 11	1 (6.25%)	1 (6.67%)	1 (6.67%)	8 (100%)
VSD, *n* = 14	6 (37.5%)	3 (20.0%)	5 (33.3%)	0
AVSD, *n* = 9	3 (18.75%)	4 (26.67%)	2 (13,3%)	0
PDA, *n* = 4	3 (18.75%)	1 (6.67%)	0	0
CCHD, *n* = 16	3 (18.75%)	6 (40.0%)	7 (46.67%)	0

ASD, atrial septal defect; AVSD, atrioventricular septal defect; PDA, persistent ductus arteriosus; VSD, ventricular septal defect; CCHD, complex congenital heart defects.

**Table 2 jcm-09-01717-t002:** Concomitant diseases in IPAH and CHD-PAH.

Disease	All Patients	IPAH	CHD-PAH	*P*
	*N* = 80	*N* = 25	*N* = 54	
Hypertension	2	2	0	0.04
Smoking	1	0	1	0.5
Asthma	2	1	1	0.57
Hypothyroidism	19	3 *	16 *	0.09
Hyperthyroidism	2	0	2	0.33
Atrial flutter	1	0	1	0.5
Down syndrome *	24	2	22	0.004
Mental retardation	32	5	27	0.01
History of acute PE	1	1	0	0.14
Inflammatory bowel disease	2	1	1	0.57
Cardiac stimulation	1	0	1	0.5
Sleep apnea	1	1	0	0.14
Preterm < 36 week	4	3	1	0.06

* Down syndrome was present in one IPAH patient with hypothyroidism and in 15 patients with CHD-PAH with hypothyroidism; PE, pulmonary embolism.

**Table 3 jcm-09-01717-t003:** Demographic and baseline characteristics of children with idiopathic pulmonary arterial hypertension (IPAH) and pulmonary arterial hypertension associated with congenital heart disease (CHD-PAH).

Characteristics	All (IPAH, CHD-PAH, PoPAH)	IPAH	CHD-PAH	*P*
	*N* = 80	*N* = 25	*N* = 54	
Female, *n* (%)	40 (50%)	12 (48%)	27 (50%)	0.9
Incident cases, *n* (%)	10 (12.5%)	9 (36%)	1 (1.9%)	<0.0001
Age at diagnosis	5.1 (2.1–8.1)	5.9 (2.5–8.3)	4.6 (2.1–7.5)	0.5
Age at enrollment	10.4 (7.9–15.2)	9.8 (7.6–11.5)	11.5 (7.9–15.4)	0.15
Functional class at diagnosis				
I	9 (11.3%)	2 (8%)	7 (12.97%)	0.5
II	17 (21.3%)	7 (28%)	9 (16.67%)	0.2
III	53 (66.3%)	16 (64%)	37 (68.51%)	0.7
IV	1 (1.3%)	0	1 (1.85%)	0.5
Functional class at enrolment, *n* (%)				
I	9 (11.3)	2 (8%)	7 (13%)	0.5
II	46 (57.5)	13 (52%)	32 (59%)	0.5
III	25 (31.3)	10 (40%)	15 (28%)	0.3
IV	0	0	0	-
RHC at diagnosis				
mPAP [mmHg]	48.0 (37.0–57.0)	48.0 (39–67)	48.0 (36–55.5)	0.54
CI [l/min/m^2^]	2.9 (1.9–3.9)	2.6 (1.9–3.6)	3.2 (2.0–3.9)	0.86
PVR [WU]	11.1 (7.2–21.3)	10.4 (6.7–22.9)	11.9 (7.7–19.6)	0.88
PVRI [WU/m^2^]	10.9 (6.5–21.1)	11.4 (7.9–26.5)	10.7 (6.4–18.5)	0.34
Clinical characterization at enrolment				
Height for age percentile	14 (2.75–50.0)	25.0 (7.0–62.0)	6.5 (2.0–44.0)	0.06
Weight for age percentile	22.0 (3.0–50)	36.0 (17.0–62.0)	13.5 (3.0–50.0)	0.08
BMI for age percentile	40.6 (8.0–70.0)	42.0 (17–62.0)	36.0 (3.0–75.0)	0.57
6 MWT [m]	429 (360–500)	450 (383–524)	420 (353–469)	0.46
NT-proBNP [ng/L]	272 (104.6–628.0)	268.7 (69.2–502.7)	290.0 (115.8–676.0)	0.38
RAA [cm^2^]	14.3 (10.0–19.0)	12.2 (8.8–14.6)	16.4 (10.4–20.4)	0.08
Anthropometric measurements in patients without genetic defects				
Height for age percentile	30.5 (5.0–62.5)	37 (8.5–64.5)	26 (3–62)	0.6
Weight for age percentile	37.5 (6.5–64.8)	39.5 (17.5–82.0)	37 (25–62)	0.4
BMI for age percentile	31.5 (3.5–64)	43.5 (13.5–72.0)	20 (3.0–55)	0.1
Anthropometric measurements in patients with genetic defects				
Height for age percentile	3 (2–9)	10 (3–25)	3 (2–7)	0.3
Weight for age percentile	7.5 (3–31)	25 (10–25)	7 (3–37)	0.5
BMI for age percentile	47.5 (16–77.5)	35 (25–30)	50 (12–80)	0.4

6 MWT, 6 min walk distance; NT-proBNP, N-terminal pro-B-type natriuretic peptide; RAA, right atrial area; mPAP, mean pulmonary artery pressure; CI, cardiac index; PVR, pulmonary vascular resistance; PVRI, pulmonary vascular resistance index.

**Table 4 jcm-09-01717-t004:** Pharmacotherapy of pediatric patients with Pulmonary Arterial Hypertension in the BNP-PL Registry at enrollment.

	Whole Group (IPAH, CHD-PAH, PoPAH)	IPAH	CHD-PAH	*P*
	*N* = 80	*N* = 25	*N* = 54	
PAH-specific therapies:				
Sildenafil [*n*,%]	57 (71%)	18 (72%)	38 (70.4%)	0.88
Tadalafil [*n*,%]	1 (1.3%)	0	1 (1.9%)	0.50
Bosentan [*n*,%]	61 (76.3%)	17 (68%)	44 (81.5%)	0.29
Treprostinil [*n*,%]	3 (3.75%)	3 (12%)	0	0.009
Epoprostenol [*n*,%]	0	0	0	-
Riociguat [*n*,%]	1 (1.3%)	1 (4.0%)	0	0.14
Ca blocker for PAH	2 (2.5%)	2 (8.0%)	0	
PAH-specific monotherapy [*n*,%]	36 (45%)	10 (40%)	24 (44.4%)	0.71
PAH-specific combination therapy of two drugs [*n*,%]	39 (49%)	10 (40%)	29 (53.7%)	0.7
Triple combination PAH-specific therapy [*n*,%]	3 (3.8%)	3 (12%)	0	0.01
Home oxygen therapy	2 (2.5%)	1 (4%)	1 (1.9%)	0.57
Vitamin K antagonists	12 (15%)	6 (24%)	6 (11.1%)	0.14
Low-molecular heparin	1 (1.3%)	1 (4%)	0	0.14
Beta blockers	6 (7.5%)	3 (12%)	3 (5.6%)	0.32
ACEI	17 (21.3%)	4 (16%)	13 (24.1%)	0.42
ARB	1 (1.3%)	0	1 (1.9%)	0.49
Loop diuretics	10 (12.5%)	3 (12%)	7 (13%)	0.9
Thiazide diuretics	1 (1.3%)	0	1 (1.9%)	0.49
Potassium-sparing diuretics	40 (50%)	15 (60%)	25 (46.3%)	0.02
SSRI	1 (1.3%)	0	1 (1.9%)	0.49
ASA	10 (12.5%)	1 (4%)	9 (16.7%)	0.12
Proton pomp inhibitors	1 (1.3%)	1 (4%)	0	0.14
Immunosuppressive drugs	1 (1.3%)	0	1 (1.9%)	0.49

ACEI, angiotensin convertase enzyme inhibitor; ASA, acetylsalicylic acid; ARB, angiotensin receptor antagonist; SSRI, serotonin reuptake inhibitor; PAH, pulmonary arterial hypertension.

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
