# Peer review of "Children and Adolescents with Pulmonary Arterial Hypertension: Baseline and Follow-Up Data from the Polish Registry of Pulmonary Hypertension (BNP-PL)"

_jcm, 2020, doi:10.3390/jcm9061717_

Round 1
Reviewer 1 Report
The study by Kwiatkowska et al. characterized 80 pediatric patients with PAH registered by the Polish registry of pulmonary hypertension providing an insight into the children with PAH in Poland. My comments are listed as follows.
- Line 49-50, please provide the references for the incidence and prevalence of PAH.
- Line 149, please clarify the prevalence in this study. Was it calculated as period prevalence? Provide the denominator used in the calculation.
- Line 263 Follow-up. Can you provide a Kaplan-Meier survival curve for the patients in this study?
- The discussion could be further improved. Are there any possible reasons causing the heterogeneous geographic distribution in PAH patients and the possible impacts on the prevalence and incidence reported?
- I think the paper would be strengthened if the findings can be compared to studies based on other PAH registries.
Reviewer 2 Report
The paper describes the prevalence and incidence of pulmonary arterial hypertension of children and adolescents in Poland.
It is not clearly described how the authors calculated the incidence rate for children (page 4). They diagnosed 10 children with PAH in 60 months. They did not mention how many childen live in Poland.
They were not able to perform genetic testing. The authors should state this clearly from the beginning. In section 3.3 page 4 they state that 25 patients were classified as IPAH. Most probably al least a few of those children carry a genetic mutation which can be linked to hereditary PAH.
In section 3.10 follow-up was described.
Are children in a bad condition, due to pulmonary hypertension, considered for screening for a lung transplantation ?
